# Intraluminal Contrast-Enhanced Ultrasonography Application in Dogs and Cats

**DOI:** 10.3390/vetsci11090443

**Published:** 2024-09-20

**Authors:** Saran Chhoey, Soyeon Kim, Eunjee Kim, Dongjae Lee, Kroesna Kang, Sath Keo, Jezie Alix Acorda, Junghee Yoon, Jihye Choi

**Affiliations:** 1Graduate School, Royal University of Agriculture, Phnom Penh 12101, Cambodia; saranchhoey@rua.edu.kh; 2Faculty of Veterinary Medicine, Royal University of Agriculture, Phnom Penh 12101, Cambodia; kkroesna@rua.edu.kh (K.K.); ksath@rua.edu.kh (S.K.); 3Department of Veterinary Medical Imaging, College of Veterinary Medicine, Seoul National University, Gwanak-ro 1, Gwanak-gu, Seoul 08826, Republic of Korea; chanchu@snu.ac.kr (S.K.); ej08ee@snu.ac.kr (E.K.); yc4537@snu.ac.kr (D.L.); heeyoon@snu.ac.kr (J.Y.); 4Department of Veterinary Medical Imaging, College of Veterinary Medicine, Chonnam National University, Yongbong-ro 77, Buk-gu, Gwangju 61186, Republic of Korea; 5College of Veterinary Medicine, University of the Philippines Los Baños, Laguna 4031, Philippines; jaacorda@up.edu.ph

**Keywords:** canine, CEUS, feline, gastrointestinal, urinary

## Abstract

**Simple Summary:**

In this retrospective study, three dogs and three cats were selected from the database of the Chonnam National University Teaching Hospital from February 2017 to June 2019. The inclusion criteria for the selection of patients were as follows: (1) clinical data including age, sex, breed, clinical signs, histology, fine-needle aspiration, or fluid centesis were available and (2) intraluminal CEUS was performed; CEUS findings helped confirm the diagnosis. Each patient underwent physical examination, blood tests, radiography, and conventional ultrasonography. Intraluminal CEUS was performed using contrast agents, including SonoVue, agitated saline, saline, or a combination for GI hydrosonography in two patients and sono-cystourethrography in four patients. GI hydrosonography could assess the anatomic relationship between the mass and gastric lumen after administration of a mixture of 0.1 mL SonoVue and 30 mL/kg water in case 1 and could dilate the colonic lumen to detect the thickened wall using saline infusion in case 2. Upon sono-cystourethrography, communication between the urinary tract and prostatic cyst was clearly visualized in case 3, and narrowing of the urethral lumen secondary to prostatomegaly could be ruled out in case 4 with pyogranulomatous prostitis using agitated saline. The saline or agitated saline is appropriate for filling the lumen and improving the acoustic window for GI hydrosonography. Intraluminal CEUS examinations helped to assess the patency of the lumen, evaluate the extent of luminal dilation, rule out luminal narrowing, determine the presence of a mass within the lumen, and identify rupture sites.

**Abstract:**

Administering intraluminal fluid can improve the acoustic window for the visualization of the lumen and wall layers in the cavitary organs. Microbubbles in ultrasound contrast agents can also be used for intracavitary applications to enhance visualization of the lesion in human patients. However, there was no literature extending the clinical application of intraluminal contrast-enhanced ultrasonography (CEUS) to patients with naturally occurring diseases in veterinary medicine. This case series aims to describe the detailed application and diagnostic value of intraluminal CEUS in six clinical cases with naturally occurring gastrointestinal (GI) and urinary tract diseases.

## 1. Introduction

Ultrasonography provides real-time evaluation of abdominal organs regarding shape, echogenicity, echotexture, margin, blood flow, and motility in veterinary medicine [1,2,3]. However, despite its extensive use, ultrasonographic assessment encounters limitations, particularly in imaging gas-filled structures, in visualizing narrow lumen structures or identifying sites of small organ rupture, attributed to suboptimal acoustic windows and limited spatial resolution [4].

To circumvent these challenges, administering intraluminal fluid can improve the acoustic window for the visualization of lumen and wall layers and enable precise measurement of the wall thickness in a fluid-dilated stomach compared to an empty or collapsed stomach [5]. In human medicine, injecting fluid into the uterine cavity assists in visualizing pathologic lesions, such as myomas, polyps, adhesions, and congenital anomalies, though it provides limited information on tubal patency [6]. Hydrosonography is a simple technique that can be easily performed in patients without the need for sedation or anesthesia. In GI hydrosonography, ultrasonography uses fluid to distend the GI tract by fluid administration and improve the acoustic window to evaluate the gastric wall and lumen.

Ultrasound contrast agents (UCAs) consist of gas-filled microbubbles encapsulated in a shell that resonate when displayed on an ultrasound beam, enhancing imaging [7,8]. Contrast-enhanced ultrasonography (CEUS) generally evaluates the perfusion of lesions after intravenous injection of UCAs [9,10,11,12,13,14,15,16,17,18]. CEUS allows for real-time evaluation of wash-in and wash-out of the contrast agent and provides hemodynamic information representative of the blood volume and flow rate in the lesion. Because microbubbles can generate hyperechoic signals by reflecting and scattering sound waves, UCAs can also be used for intracavitary applications as an intraluminal CEUS to enhance the visualization of abscesses or fistulas in human patients [19,20,21,22]. For instance, the sulfur-hexafluoride SonoVue^®^ (Braco, Milan, Italy) has been used to guide abscess drainage by being injected via drainage catheters, while hydrogen peroxide has helped identify anal fistulas and their internal openings by injection into the fistula tract [23,24,25,26,27,28].

Despite its proven benefits in human medicine, there have been only four reports of intraluminal CEUS use in veterinary medicine [28,29,30,31,32]. Traditionally, 30 mL/kg of tap water has been used for GI hydrosonography. In a previous study, GI hydrosonography using SonoVue for the assessment of the GI tract was evaluated by comparing non-contrast ultrasonography and hydrosonography using water in healthy beagle dogs [28]. After intraluminal administration of 0.1 mL of SonoVue mixed with 30 mL/kg of water into the stomach, wall thickness, wall layer definition, conspicuity of the mucosal surface, and image quality were evaluated in the gastric cardia, body, pylorus, and cranial part of the descending duodenum [29]. GI hydrosonography using SonoVue significantly improved the wall definition and mucosal conspicuity in the near fields of the gastric body, pylorus, and duodenum compared to the non-contrast method and the pylorus compared to the water method [29,33].

Sono-urethrography, another application, assesses urethral integrity through fluid dilation of the urethral lumen. In a previous study, sono-urethrography using saline, agitated saline, and SonoVue was performed to evaluate urethral integrity by fluid dilation of the urethral lumen and urethral wall thickening in male dogs [28] and humans [34]. In sono-urethrography, saline and agitated saline are recommended for evaluation of the urethral wall and lumen and an UCA is recommended for evaluation of the integrity of the urethra by improving its conspicuity. In another study, contrast-enhanced voiding ultrasonography was performed after injecting a gently shaken mixture of 1 mL of SonoVue and 250 mL of 0.9% saline through a urinary catheter into the urinary bladder and urethra in healthy dogs [31]. Contrast-enhanced cystosonography using 1 mL/kg agitated saline, predetermined to contain 20% air by volume, was used to detect urinary bladder ruptures in 17 dog cadavers [32].

The majority of these studies assessed the feasibility of intraluminal CEUS in healthy dogs; however, there was no literature extending its clinical application to patients with naturally occurring diseases. Therefore, in this study, the intraluminal CEUS features were described in dogs and cats that underwent intraluminal CEUS examinations of the GI tract and lower urinary tract. This study aimed to introduce the clinical application of intraluminal CEUS in dogs and cats.

## 2. Case Description

This case series included three dogs and three cats that underwent intraluminal CEUS using various contrast agents, including SonoVue, agitated saline, and saline, or a combination thereof. Two patients received GI hydrosonography and four patients underwent sono-cystourethrography. All intraluminal CEUS examinations were conducted using an ultrasound machine (Prosound Alpha 7, Hitachi-Aloka, Tokyo, Japan) with a 10 MHz linear transducer by a radiologist (J.H.C.). The time gain control was set at its central position and the power was set at a constant level, with a gain of 64 dB.

### 2.1. Case 1

A 12-year-old spayed female Maltese presented with vomiting, anorexia, and polydipsia for two weeks. The patient showed mild anemia, increased blood urea nitrogen (>130 mg/dL, reference range; 7–27 mg/dL) and creatinine (>2.7 mg/dL, reference range; 0.5–1.8 mg/dL) levels, hyperphosphatemia (14.6 mg/dL, reference range; 2.5–6.8 mg/dL), and hypokalemia (2.5 mmol/L, reference range; 3.5–5.8 mmol/L). The urine-specific gravity was 1.014. Ultrasonography revealed a hyperechoic change of the bilateral renal cortex with attenuation of corticomedullary definition, leading to a diagnosis of chronic renal disease. A crater-like mucosal defect filled with hyperechoic gas was identified in the gastric wall at the pylorus (Figure 1). The adjacent gastric wall appeared uniformly thickened with an irregular mucosal surface and poorly defined wall layers. Additionally, a focal hypoechoic-thickened lesion measuring 9.6 mm in thickness was observed. Due to the stomach being mostly contracted and filled with a significant amount of gas, it was difficult to accurately assess the thickness of the gastric wall and the extent of the lesion, and it was challenging to determine whether the layers of the stomach wall were lost. Therefore, GI hydrosonography was conducted with the oral administration of 100 mL of saline. The saline expanded the stomach and reduced the degradation of the quality of the ultrasound images caused by gas effects. The mucosal defect was measured 16.9 mm in length and located in the gastric body. Gas accumulation within the gastric wall appeared as an echogenic line. In the dilated pylorus, another crater-like lesion was clearly visible, which had a thinner wall thickness (2.6 mm) compared to the adjacent normal layer thickness (5.9 mm). The focal thickened area of the pyloric wall was measured at 25 × 6.7 mm, with a loss of clear definition in the wall layers. The gastric wall thickening was considered to be uremic gastropathy induced by chronic renal failure. The azotemia was alleviated over 14 days with medication; however, the dog was euthanized due to the grave prognosis of chronic renal failure. The gastric lesion was determined to be a moderate multifocal erosion histopathologically.

### 2.2. Case 2

A 6-month-old Scottish Fold cat, diagnosed with feline infectious peritonitis, presented with intermittent vomiting, diarrhea, and constipation over the course of a week. Conventional ultrasonography revealed a small amount of anechoic fluid, generalized swollen abdominal fat, moderately enlarged lymph nodes, and hyperechoic renal cortex with medullary rim sign in both kidneys. The wall thickness and layers of the ascending and transverse colons appeared normal (Figure 2). However, the descending colon showed marked wall thickening, measuring 10 mm in thickness over a length of approximately 70 mm, and fecal retention was observed immediately proximal to the thickened area of the colon. In particular, the thickening of the muscular layer was considered; however, luminal obstruction and the definition of the wall layers could not be confirmed because the colon collapsed, which could make the wall appear thick. GI hydrosonography was performed by administering 10 mg of tap water via a rectal catheter to assess the extent of the thickened descending colon and evaluate the colonic wall. The distal part of the descending colon, which appeared thickened and corrugated in conventional ultrasonography, was dilated normally and showed a regular mucosal surface and thin walls. However, the proximal two-thirds of the descending colon did not show expansion of the lumen due to hypertrophied bowel walls, and all layers of the bowel wall appeared to be preserved normally. However, the muscular layer was markedly hypertrophied, suggesting inflammatory changes related to feline infectious peritonitis. Compared to conventional ultrasonography, the use of GI hydrosonography provided clearer assessment of the length of the hypertrophied bowel walls and evaluation of the wall layers.

### 2.3. Case 3

An 11-year-old intact male Pekingese presented with dark brown or bloody hematuria and decreased appetite for 15 days. Physical and laboratory examinations were within normal ranges, including serum creatinine (0.9 mg/dL; reference range 0.5–1.8 mg/dL) and blood urea nitrogen (18 mg/dL; reference range 7–27 mg/dL) levels. Conventional ultrasonography revealed an asymmetrically enlarged prostate. The right lobe was replaced by a cystic lesion (46.27 × 31.86 mm) containing hyperechoic fluid with loss of most of the prostatic parenchyma (Figure 3). The prostatic part of the urethra was dilated and displaced to the left by the enlarged right prostate lobe. The urinary bladder and urethra contained hyperechoic sludge, and the bladder wall was thickened. The peri-prostatic fat was normal; however, the medial iliac lymph nodes were enlarged to approximately 12 mm, with an oval shape. Because the hypoechoic fluid mixed with hyperechoic materials, suggestive of infected sludge found in the urinary bladder and prostatic cyst, communication between the urinary tract and the prostatic cyst was suspected but not confirmed using conventional ultrasonography. Therefore, sono-urethrography was performed by injecting agitated saline into the urethra via a 6-French urinary catheter that was passed into the distalpenile part of the urethra. After connecting two syringes, one containing 20 mL of 0.9% saline and the other empty, to a three-way stopcock, agitated saline was prepared by the rapid injection of saline from the full syringe into the empty syringe several times. Then, the agitated saline was immediately injected into the urethral catheter. The hyperechoic bubbles flowed into the urinary bladder, and some of them leaked into the prostatic cyst. Communication between the urinary bladder and the prostatic cyst was clearly observed because the saline bubbles showed hyperechoic signals in contrast to the hypoechoic fluid in the cyst. Under ultrasound guidance, approximately 20 mL of turbid, creamy, yellow fluid was obtained from the prostatic cyst using a 23-gauge catheter. The sampled fluid was analyzed; the total nuclear cell count was 55.33 K/μL, the specific gravity was 1.020, the creatinine level was 48.8 mg/dL, the fluid/serum creatinine ratio was 54.2, and the total protein level was less than 2.0 g/dL. The fluid contained degenerated neutrophils and rod-shaped bacteria. Streptococcus and Pasteurella were detected in both fluid and urine cultures. Benign prostatic hyperplasia, prostatic urethral leakage, a prostatic cyst, and subsequent lower urinary tract infection were confirmed. Castration was performed, and antibiotics were administered.

### 2.4. Case 4

A 12-year-old intact male Shih Tzu was presented with symptoms of dysuria, anorexia, and depression. The dog was diagnosed with urethral calculi approximately 2 years before and recently experienced pain during urination. Clinical examination and findings upon complete blood count and serum biochemistry evaluation were unremarkable. Upon conventional ultrasonography, a calculus, approximately 3.5 mm in diameter, was visible in the penile urethra; however, there was no dilation of the urethral lumen. The prostate was enlarged to approximately 20 × 23 mm and showed a mildly heterogeneous echotexture with hypoechoic areas (Figure 4). It had regular margins, and there was no cyst or mineralization within the prostatic parenchyma. The prostatic part of the urethra was not clearly observed, and it was difficult to confirm patency or the obstruction of the urethral lumen. The medial iliac lymph nodes were mildly enlarged to 1.4 cm in length, but their shapes were normal. The prostatomegaly was attributed to benign prostatic hyperplasia with the likelihood of a prostatic tumor considered low because the dog was a neutered male. To evaluate the prostate parenchyma, CEUS was performed after the intravenous injection of 0.125 mL/kg SonoVue, followed by a bolus of 5 mL saline through the cephalic vein. This examination was performed in the extended pure harmonic detection mode with the focus point set at the bottom for homogeneous energy distribution over the image. The transmitted energy was reduced to magnitudes of 7%, and the MI used was 0.07. In the wash-in phase, both the prostatic lobes were simultaneously enhanced; however, the left lobe showed a faster washout than the right lobe. Sono-urethrography by injecting agitated saline via a 6-French urinary catheter that was passed into the penile part of the urethra was performed to evaluate the narrowing of the urethral lumen secondary to prostatomegaly and to determine the cause of the dysuria. Microbubbles passing through the intact lumen of the prostatic urethra were clearly visible, and there was no evidence of abnormal changes in the urethral mucosa. Urethral compression or obstruction was ruled out. Fine-needle aspiration of the left prostatic lobe revealed pyogranulomatous inflammation of the prostatic parenchyma.

### 2.5. Case 5

A 3-year-old neutered male Russian Blue who developed acute ascites following cystocentesis performed for feline idiopathic cystitis at a local clinic. Laboratory tests revealed the elevation of the serum creatinine (3.7 mg/dL; reference range 0.8–2.4) and lipase (168 U/L; reference range 0–35 U/L) levels and mild hyponatremia (139.5 mmol/L; reference range 146–157 mmol/L) and hypochloremia (113.6 mmol/L; reference range 116–126 mmol/L). Upon conventional ultrasonography, a large amount of peritoneal fluid surrounded the urinary bladder. A hyperechoic line was observed within the bladder wall, which was suspected to be the ruptured area. Sonocystography was performed to determine the ruptured area (Figure 5). Initially, agitated saline was administered via a urethral catheter, and the bladder was distended normally without significant leakage of bubbles into the peritoneal cavity. Sono-cystography using SonoVue was conducted by injecting 0.5 mL (25 mg) of the UCA, followed by a 5 mL saline flush. The strong bright signal of the microbubbles indicated leakage from the suspected ruptured area of the bladder into the peritoneal cavity. The bladder rupture was treated by keeping a urinary catheter in place for approximately 3 days instead of performing a laparotomy because the ruptured area was very small. Sono-urethrography showed a small defect in the urinary bladder wall in the previously ruptured region; however, there was no evidence of bubble leakage.

### 2.6. Case 6

A 2-year-old male mixed cat presented with symptoms of depression and abdominal distension, suggestive of urinary bladder rupture following trauma. Upon conventional ultrasonography, the urinary bladder was empty and collapsed to a thickness of 8 mm of the bladder wall. Cystic lesions (28.6 × 20 mm) containing echogenic fluid were observed on the right side of the bladder apex. The cystic structures appeared to be located within the bladder wall without a clear distinction from the remaining bladder. Large amounts of anechoic peritoneal effusion and hyperechoic swelling of fat were observed in the caudal abdomen. Approximately 470 mL of the effusion was removed via abdominal centesis. Sono-urethrography was additionally performed to demonstrate the anatomic relationship between the cystic structure and the urinary bladder and to evaluate their integrities (Figure 6). When agitated saline was used, echogenic bubbles filled the bladder; however, bladder rupture was not identified. However, when the sono-urethrography was repeated using SonoVue, the contrast agent flowed through the bladder and the cystic lesion. Echogenic linear leakage of the microbubbles was clearly observed from the cystic lesion, which flowed into the abdominal cavity. Laparotomy confirmed a bladder defect at the apex with a hematoma.

## 3. Discussion

This retrospective study introduced the clinical application of intraluminal CEUS for GI and urinary tracts in dogs and cats. Intraluminal CEUS in GI hydrosonography and sono-urethrography improved the visualization of lesions in naturally occurring diseases.

In this present study, GI hydrosonography was performed in the upper GI tract and colon. Traditionally, GI hydrosonography is performed using tap water [5,29]. In human medicine, antifoaming agents such as simethicone and microsphere UCAs such as SonoVue are used for GI hydrosonography [5,35,36]. In this present study, a mixture of SonoVue and water was used as the contrast agent for GI hydrosonography of the stomach rather than using water only. A sufficient amount of anechoic water improved the acoustic window by filling the stomach, and hyperechoic microbubbles clearly enabled visualization of the lumen extending from the gastric fundus to the pylorus by enhancing the mucosa–lumen interface. This result is comparable to those of previous studies performed on healthy dogs and human patients [5,29,35]. GI hydrosonography using a SonoVue and water mixture in healthy dogs showed the homogeneous echogenicity of the luminal border and improved wall layer definition and mucosal conspicuity of the GI tract compared with non-contrast studies or GI hydrosonography using water only in healthy dogs [29]. The dilution ratio of UCA to water may affect the image quality. In human studies, approximately 0.05 to 0.1 mL of SonoVue was mixed with 200 mL of tap water. However, this present study was conducted using the same dilution ratio and volume of the UCA used in a previous healthy canine study since the study provided image quality good enough to evaluate the wall and lumen of the GI tract [29].

In case 2, we used only water for hydrosonography of the colon to differentiate true wall thickening from pseudo-thickening caused by the collapse of an empty colon. Colonic wall thickness can be affected by luminal contents [37]. When the colon was evaluated after rectal administration of three different contrast agents, including tap water, diluted barium, or air, on ultrasonography in healthy dogs, the thickness of the colonic wall was highest after the administration of water, followed by barium and air [37]. In addition, hydrosonography using water alone allows visualization of the mucosal and muscular layers of the colon [37]. Similarly, in humans, the infusion of saline decreased the concentration of fecal residues and improved the clarity of the colonic wall [38]. In 17 children, GI hydrosonography of the colon using saline allowed for the evaluation of the entire colon and detection of colorectal polyps [38]. Moreover, the administration of warm tap water into the rectum can increase the diagnostic accuracy of pelvic masses and delineate the extent of tumors arising from pelvic organs such as the bladder, prostate, and uterus by improving the acoustic window in human patients [33].

Saline infusion is commonly used in sono-urethrography in humans [36,39]. Sono-urethrography can show various imaging features and has several strengths depending on the UCA used [28]. In healthy dogs, agitated saline and SonoVue showed hyperechoic signals created by bubbles, which were useful for the identification of the urethra compared to anechoic saline [28]. Sono-urethrography using saline or agitated saline infusion could improve the visualization of the urethral wall and lumen because it does not induce significant shadowing artifacts, which can impair the urinary tract image quality. In our study, sono-urethrography was performed using agitated saline in all four cases because we expected that the hyperechoic signals generated from the bubbles would improve urine flow in dogs with urinary tract abnormalities. The hyperechoic bubbles could be formed easily in agitated saline, allowing visualization of the urethral lumen and communication between the prostatic cysts because the saline bubbles showed hyperechoic signals in contrast with the hypoechoic fluid in the cyst and urethral lumen. Moreover, agitated saline clearly showed the normally dilated lumen of the prostatic urethra and excluded narrowing or obstruction of the urethra by the enlarged prostatic lobe in dogs with pyogranulomatous prostitis. Similarly, sono-urethrography using saline or UCAs has been used to detect anterior urethral strictures and to clarify their location and length in human patients [36,39].

Microbubbles in agitated saline and UCAs can improve image quality by absorbing, attenuating, and scattering ultrasound signals more than the surrounding tissue, subsequently enhancing the image contrast of microbubbles from the surrounding tissues [40,41]. However, our study found that sono-urethrography using agitated saline was limited in detecting and localizing bladder rupture, and required additional sono-urethrography using SonoVue. The use of SonoVue in sono-urethrography generated a stronger and brighter signal of microbubbles than agitated saline. This allowed us to successfully demonstrate leakage from small, ruptured areas of the bladder into the peritoneal space. A previous study involving healthy dogs indicated the potential of sono-urethrography using UCA to assess urethral integrity, and this was supported by improved conspicuity of the urethra when using UCAs as opposed to saline or agitated saline [28]. This result could be related to the different characteristics of the microbubbles in agitated saline and UCAs, such as size, gas composition, solubility of air in the fluid, and stability [40,42]. Microbubbles in the agitated saline are generally considered >10 μm in diameter and larger than the microbubbles in UCAs at approximately 1–10 micrometer diameters [42]. Microbubbles in agitated saline are unstable and rapidly dissolve because of their high solubility compared to UCA microbubbles, which contain fluorinated gases, such as sulfur hexafluoride, and are stabilized by phospholipids and other amphiphilic polymeric shells [42]. Therefore, the UCA induced a strong and uniform signal that was strong enough for the detection of hyperechoic microbubbles leaked from small defects in the urinary tract.

Although saline infusion was not used to evaluate the urinary tract in our study, its use in sono-urethrography should be limited to dogs with urinary diseases because of the short duration of the examination with saline infusion in healthy dogs [28]. Considering the need to scrutinize the entire urethra in patients, the continuous infusion of saline may be required to prevent urethral collapse that can occur after the completion of saline infusion.

In this study, sono-urethrography was applied to our cases as an alternative to traditional radiographic contrast studies such as retrograde urethrography and voiding cystourethrography, which are commonly used in veterinary medicine to evaluate the diameter, wall, lumen, and integrity of the urethra [43]. The reason for this choice was the limited capability of radiographic urethrography in detecting subtle mucosal changes within the urethra. Additionally, sono-urethrography could be easily performed on the lower urinary tract without the need for sedation or anesthesia. This technique provides real-time imaging of the urethra in females and the prostatic as well as penile urethra in male dogs and facilitates repeated examinations until the lesion’s features are clearly assessed [36,44].

This study had several limitations. First, intraluminal CEUS was performed only as GI hydrosonography and sono-urethrography because this study was performed in patients with naturally developed conditions. Second, sonographic examination of urinary conditions was conducted in only a small number of patients. Further studies on the application of this technique in various urinary tract conditions, including tumors, inflammation, and strictures, which can induce narrowing or obstruction of the urethral lumen, filling defects, and thickening and irregularity of the mucosa, are needed in dogs and cats.

## 4. Conclusions

In conclusion, intraluminal CEUS examinations were performed using various UCAs in patients with GI and urinary diseases. Intraluminal CEUS examinations helped to assess the patency of the lumen, evaluate the extent of luminal dilation, rule out luminal narrowing, determine the presence of a mass within the lumen, and identify rupture sites. Intraluminal CEUS using saline or agitated saline is appropriate for filling the lumen and improving the acoustic window for GI hydrosonography. Intraluminal CEUS with agitated saline and SonoVue is useful for assessing abnormal connections and detecting ruptures. Intraluminal CEUS using SonoVue, water, agitated saline, and saline is useful and easily performed to evaluate the GI tract, lower urinary tract, and prostate in dogs and cats without the need for sedation or anesthesia.

## Figures and Tables

**Figure 1 vetsci-11-00443-f001:**
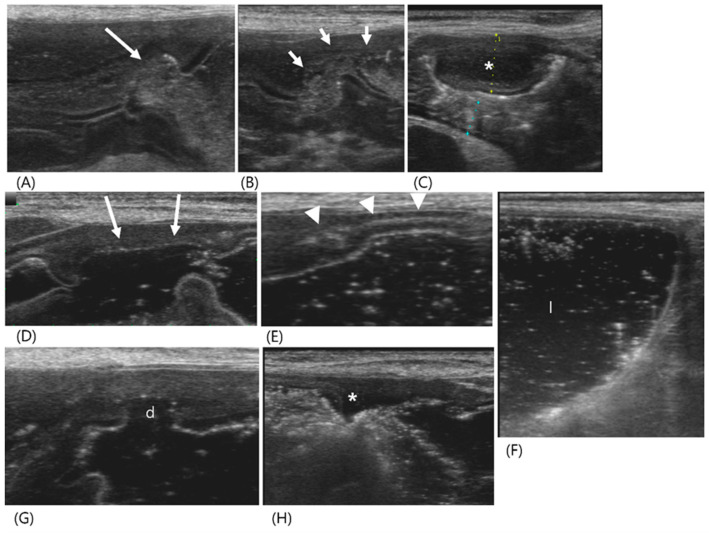
Ultrasonography of a 12-year-old spayed female maltese (Case 1) with uremic gastropathy before (**A**–**C**) and after GI hydrosonography (**D**–**H**). Upon conventional ultrasonography, (**A**) a crater-like mucosal defect (long arrow) was filled with hyperechoic gas in the gastric wall at the pylorus. (**B**) The collapsed pyloric antrum (short arrows) had a uniformly thickened wall with an irregular mucosal surface and poorly defined layers. (**C**) A focal hypoechoic wall thickening (asterisk) was shown. (**D**–**H**) After oral administration of saline, the gastric lumen was filled with fluid. (**D**) The mucosal defect (long arrows) was larger than that in (**A**) and located in the gastric body but not in the pylorus. (**E**) Note the echogenic line (arrowheads) within the gastric wall generated by intramural gas accumulation. (**F**) Normal wall thickness and layers were detected by the dilated gastric lumen (l). (**G**,**H**) The pylorus had irregular wall thickness due to another crater-like lesion (d), clearly thinner compared to the adjacent wall and focal thickened area (asterisk). Note a loss of clear definition in the wall layers in the focal thickened area.

**Figure 2 vetsci-11-00443-f002:**
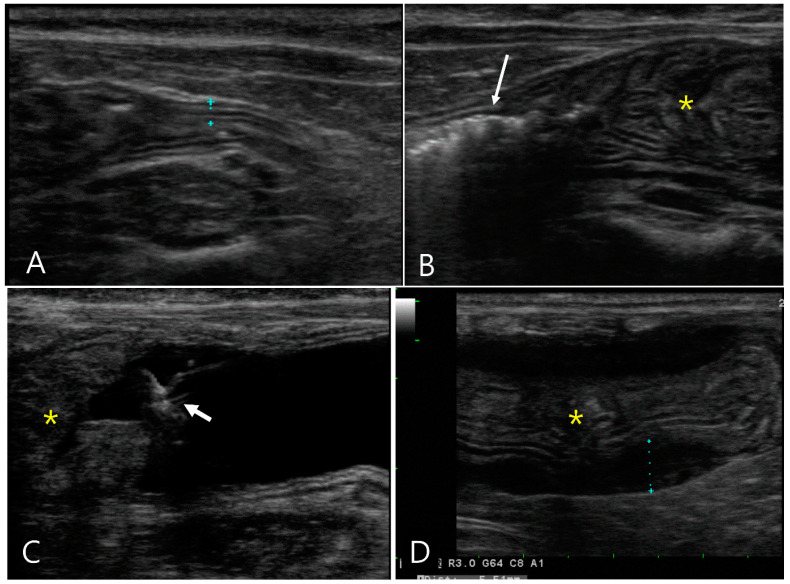
Longitudinal images of abdominal ultrasonography in a 6-month-old Scottish fold cat diagnosed with feline infectious peritonitis (Case 2) pre- (**A**,**B**) and post-GI hydrosonography (**C**,**D**). Upon conventional ultrasonography, (**A**) the ascending colon had a normal wall thickness (1.96 mm) with distinct layer delineation. (**B**) The descending colon (asterisk) exhibits marked thickening, measuring 1 cm in thickness and extending over 7 cm in length, with collapsed lumen. Note the presence of gas and feces (long arrow) proximal to the affected area. (**C**) Following saline infusion via a rectal catheter (short arrow), the distal descending colon was normally dilated with anechoic fluid (f) visible. The thickened area of the colon (asterisk) was not fully dilated. (**D**) The proximal two-thirds of the descending colon (asterisk) showed no lumen expansion; the muscular layer (dotted line) was markedly hypertrophied, although the wall layers remain normally preserved.

**Figure 3 vetsci-11-00443-f003:**
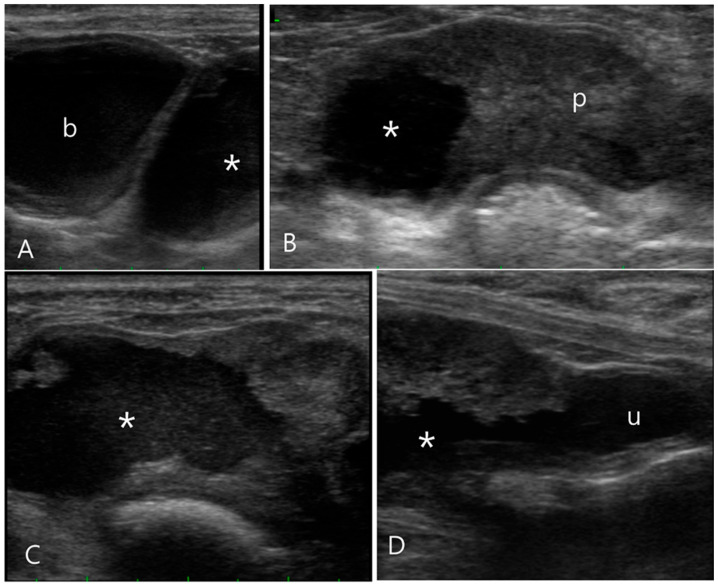
Ultrasonography of the prostatic cyst in a 11-year-old intact male Pekingese (Case 3). (**A**) Upon longitudinal ultrasound image, the asymmetrically enlarged prostate with a cystic formation (asterisk) was observed caudal to the bladder (b). (**B**) Transverse ultrasound images revealed the right prostate lobe almost entirely replaced by a cystic lesion filled with hyperechoic fluid, unlike the left prostatic lobe (p). (**C**,**D**) Sono-urethrography images obtained using agitated saline to enhance contrast, displaying the presence of microbubbles indicating the echogenic fluid leakage into the prostatic cyst. Additionally, these images demonstrate a clear communication between the prostatic cyst and urethra (u).

**Figure 4 vetsci-11-00443-f004:**
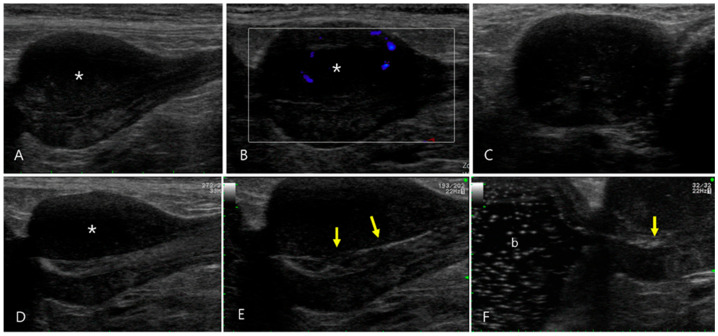
Ultrasonography of pyogranulomatous inflammation of the prostate in a 12-year-old intact male Shih Tzu (Case 4). (**A**) Prostatomegaly characterized by a regular margin and mildly heterogeneous echotexture with notable hypoechoic changes in the parenchyma of the left lobe (asterisk) was observed. (**B**) Color flow Doppler images revealed minimal vascular signal shown as blue signals in the prostate, suggesting low blood flow. (**C**–**E**) Sono-urethrography conducted with agitated saline showed microbubbles transiting through the intact lumen of the prostatic urethra, highlighting the regular appearance of the urethral mucosal surface (arrows). (**F**) The echogenic bubbles flowed into the urinary bladder (b), confirming the absence of urethral compression or obstruction.

**Figure 5 vetsci-11-00443-f005:**
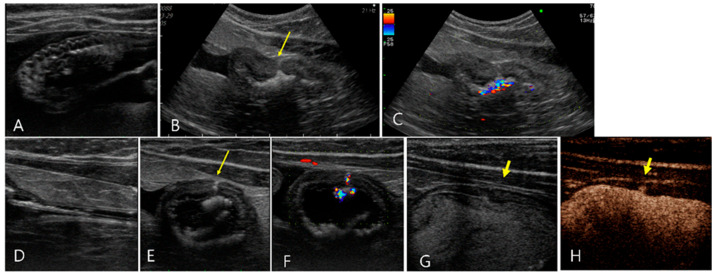
Ultrasonography of a 3-year-old neutered male Russian Blue (Case 5) presenting with urinary bladder rupture initially observed on day 1 (**A**–**F**) and subsequent findings 3 days post-catheterization (**G**,**H**). (**A**,**B**) Initial conventional ultrasonography revealed a large amount of peritoneal fluid surrounding the urinary bladder and a hyperechoic line (long arrow) indicative of a tear on the ventral aspect of the bladder wall. There were echogenic contents with acoustic shadowing in the bladder. (**C**) Note the color Doppler signals emanating from these echogenic contents, suggestive of the presence of gas or calculi. (**D**–**F**) Sono-urethrography using SonoVue demonstrated the strong bright signal of microbubbles indicating an intact urethra, with notable leakage (long arrow) of these bubbles from the damaged bladder wall into the peritoneal cavity, confirming bladder rupture. (**F**) Note the color signal within the bladder wall, attributed to microbubble presence. Follow-up sono-urethrography conducted 3 days after bladder catheterization showed a small defect (short arrow) in the bladder wall but no evident microbubble leakage, either upon conventional B-mode (**G**) or harmonic mode (**H**) imaging, suggesting partial healing of the bladder wall.

**Figure 6 vetsci-11-00443-f006:**
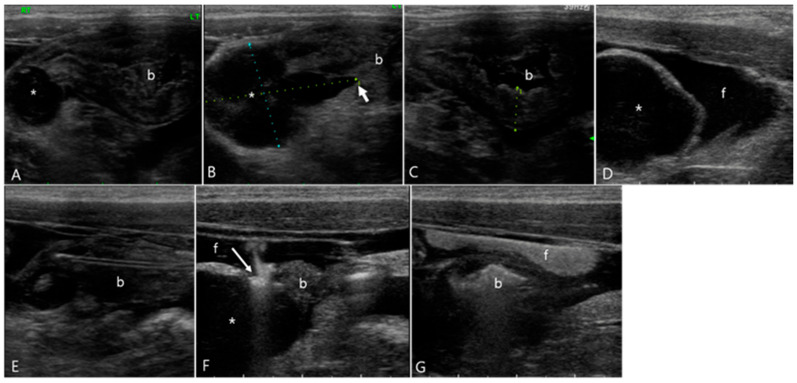
Ultrasonography of urinary bladder rupture in a 2-year-old male mixed cat (Case 6). (**A**,**B**) Cystic lesions (asterisk) filled with echogenic fluid were found within the bladder wall, located at the right side of the bladder apex. (**C**) The urinary bladder (b) was collapsed with an empty lumen and a bladder wall thickness of 8 mm. (**D**) Extensive anechoic peritoneal effusion (f) surrounded the cystic lesion (asterisk). (**E**–**G**) Upon sono-urethrography using SonoVue, a distinct echogenic linear leakage of the microbubbles (arrow) emanated from the cystic lesion, dispersed into the surrounding anechoic peritoneal effusion (f). (**E**) The urinary catheter was visualized as echogenic double lines in the bladder. (**G**) Note the change of echogenicity of the peritoneal effusion after sono-urethrography, indicating the diffusion of SonoVue microbubbles.

## Data Availability

Dataset is available upon request from the author.

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
