# Peer review of "Intraluminal Contrast-Enhanced Ultrasonography Application in Dogs and Cats"

_vetsci, 2024, doi:10.3390/vetsci11090443_

Round 1
Reviewer 1 Report
Comments and Suggestions for Authors
1A small sample of subjects was used, and different organ systems were studied.
2. Was sedation used for injecting contrast fluid into different organ systems lumen ?
Is this test method technically easy (no sedation required) and cheap enough to be proposed for routine testing?
What complications have been observed and what complications are theoretically possible?
How should the patient be prepared for the examination?
Although it is written that for the assessment of the gastrointestinal tract, lower urinary tract and prostate of dogs, cats, the examination is done without anesthesia, does it really not cause discomfort and stress to the animal?
The photos could perhaps have been presented with a more uniform design for all cases?
Author Response
- A small sample of subjects was used, and different organ systems were studied.
>> You’re right. That’s why we presented this manuscript as a case report.
- Was sedation used for injecting contrast fluid into different organ systems lumen ? Is this test method technically easy (no sedation required) and cheap enough to be proposed for routine testing?
>> Right. To reduce the confusion, I clarified the sentences like ‘Intraluminal CEUS using SonoVue, water, agitated saline, and saline is useful and easily performed to evaluate the GI tract, lower urinary tract, and prostate in dogs and cats without the need for sedation or anesthesia.’
- What complications have been observed and what complications are theoretically possible?
>> There were no complications related to CEUS in any of the animals. Theoretically, gas bubbles can induce thrombosis if they enter the bloodstream and are large enough. However, since SonoVue contains microbubbles that have been proven safe for intravenous injection, the likelihood of causing complications such as thrombosis is extremely rare.
- How should the patient be prepared for the examination? Although it is written that for the assessment of the gastrointestinal tract, lower urinary tract and prostate of dogs, cats, the examination is done without anesthesia, does it really not cause discomfort and stress to the animal?
>> This examination involves evaluating with ultrasound after injecting contrast agents such as ultrasound contrast agents or saline, instead of performing radiography following the injection of iodine contrast agents. Just as no significant side effects have been commonly reported during radiographic contrast study with iodine contrast agents, the likelihood of adverse reactions occurring during a CEUS examination is also very low. Therefore, patient preparation only requires careful attention to catheter placement for the injection of the contrast agent, without any need for special preparation.
- The photos could perhaps have been presented with a more uniform design for all cases?
>> That's a good point. However, since the examination areas and the CEUS findings to emphasize are different in each case, we had no choice but to present a variety of images. We believe that presenting the images this way will help in understanding the patient's examination process and findings.
Reviewer 2 Report
Comments and Suggestions for Authors
The reviewer manuscript brings a valuable information for the sonographic imaging technique and its accuracy development. The intraluminar CEUS, frequently used in human medicine, is still rarity in veterinary diagnostic.
The work is well organized and written. Each case report gives full information about animal status praesens, symptoms, diagnosis and treatment. The conventional sonography is compared with the CEUS, which brings new important information (lesions existence, morphology etc.).
The animals used in the study were the object of standard veterinary care. The sonoimages have good quality and description.
I have added some minor remarks to the pdf file of the manuscript. I hope they will be helpful.
I suggest to accept reviewed paper after minor revision.

Author Response
The reviewer manuscript brings a valuable information for the sonographic imaging technique and its accuracy development. The intraluminar CEUS, frequently used in human medicine, is still rarity in veterinary diagnostic.
The work is well organized and written. Each case report gives full information about animal status praesens, symptoms, diagnosis and treatment. The conventional sonography is compared with the CEUS, which brings new important information (lesions existence, morphology etc.). The animals used in the study were the object of standard veterinary care. The sonoimages have good quality and description. I have added some minor remarks to the pdf file of the manuscript. I hope they will be helpful. I suggest to accept reviewed paper after minor revision.
>> Thank you for your detailed correction suggestions. I have revised the content of the manuscript according to the comments you provided.